# Caring for the Caregivers: Improving Mental Health among Health Professionals Using the Behavioral Health Professional Workforce Resilience ECHO Program

**DOI:** 10.3390/healthcare12171741

**Published:** 2024-08-31

**Authors:** Jeffrey W. Katzman, Laura E. Tomedi, Navin Pandey, Kimble Richardson, Stephen N. Xenakis, Sarah Heines, Linda Grabbe, Yasmin Magdaleno, Ankit Mehta, Randon Welton, Kelly Lister, Kelly Seis, Antoinette Wright, Shannon McCoy-Hayes, Joanna G. Katzman

**Affiliations:** 1Project ECHO, University of New Mexico Health Sciences Center, Albuquerque, NM 87106, USA; 2Department of Psychiatry and Behavioral Sciences, School of Medicine, University of New Mexico Health Sciences Center, Albuquerque, NM 87106, USA; 3College of Population Health, University of New Mexico Health Sciences Center, Albuquerque, NM 87106, USA; 4Community Health Network—Behavioral Health, Indianapolis, IN 46256, USA; 5Silver Hill Hospital, New Caanan, CT 06840, USA; 6Nell Hodgson Woodruff School of Nursing, Emory University, Atlanta, GA 30322, USA; lgrabbe@emory.edu; 7Department of Pediatrics, School of Medicine, University of New Mexico Health Sciences Center, Albuquerque, NM 87106, USA; ymagdaleno@salud.unm.edu; 8Department of Internal Medicine, University of Minnesota, Minneapolis, MN 55455, USA; 9Department of Psychiatry, Northeast Ohio Medical University, Rootstown, OH 44272, USA; 10Department of Neurosurgery, School of Medicine, University of New Mexico Health Sciences Center, Albuquerque, NM 87106, USA

**Keywords:** resilience, health care worker well-being, mental health, tele-mentoring, burnout, self-care

## Abstract

Behavioral health professionals are at high risk for burnout and poor mental health. Our objective was to understand the impact of the Behavioral Health Providers Workforce Resiliency (BHPWR) ECHO Program on the resilience and burnout of participating behavioral health professionals. We assessed the first two years (March 2022 to March 2024) of the BHPWR ECHO, a national program operating from the University of New Mexico (N = 1585 attendees), using a mixed-methods design. We used a retrospective pre/post survey (n = 53 respondents) and focus interviews with 1–3 participants (n = 9 participants) to assess for changes in knowledge and confidence and assess changes in burnout and resilience. We found that participants increased their knowledge of how to respond when (1) their workload was more than they could manage, (2) they felt that they lacked control, (3) their work did not feel rewarding, and (4) they were experiencing compassion fatigue. They increased their confidence in (1) building a support system and (2) using the wellness tools taught in the course. Respondents were less burnt out (score: 26.0 versus 17.8, *p* < 0.01) and more resilient (29.9 versus 34.9, *p* < 0.01) compared to when they started attending the program. Tele-mentoring programs like the BHPWR ECHO Program may improve wellness among health care professionals.

## 1. Introduction

Burnout among health care professionals providing behavioral health care is a major concern for the health care system. Burnout is characterized as the state of mental and physical exhaustion related to work or caregiving activities and can be defined as a state of emotional exhaustion, depersonalization, and low personal accomplishment [1]. Professionals across the health care workforce spectrum have been at significant risk for the development of burnout, compassion fatigue, isolation, and other mental health experiences, including substance use/abuse, post-traumatic stress disorder (PTSD), depression, and suicide [2]. Estimates of burnout among family practice physicians approximate 60% [3], and 20–60% of medical specialists describe symptoms of burnout [4]. Burnout in the health care system has been associated with excessive workload, family/work conflicts, clerical issues, expansion of the electronic medical record [5], and inefficient systems of care. Approximately one in three physicians and one in eight dentists experience burnout at any given time [2,6,7,8]. Unfortunately, the COVID-19 pandemic and associated grief, trauma, isolation, and depression have exacerbated burnout among the entire health care workforce. Intent to leave increased to 40% with decreasing job satisfaction across health care disciplines [9].

Behavioral health providers are particularly susceptible to burnout. The pandemic has worsened the mental health status of Americans, leading to greater demands on behavioral health providers’ time, an increased caseload, and reduced time for self-care. As many as 67% of mental health providers—including psychiatrists, social workers and counselors, and psychiatric nurses—have reported high levels of burnout, including emotional exhaustion, compassion fatigue, and depression [1,10]. Seventy-eight percent of psychiatrists report some of these symptoms [11].

Stress and potential burnout extend across the lifespan of those involved in providing health care, beginning with experiences in medical education. It is predicted that over half of teaching faculty will retire by 2026—raising great concern for the National Conference of Deans [12]. Burnout is common within the continuum of medical training and practice and is associated with depression, loneliness, suicidal ideation, and lower patient satisfaction and perceived quality of care [12,13]. Time demands, lack of control, work planning, work organization, difficult job situations, and interpersonal relationships are factors contributing to trainee burnout. Burnout has been shown to be problematic in psychiatric residency programs. A recent review of 22 studies looking at this issue identified a burnout rate for psychiatry residents of 37% [14].

Given the increased stress for health care professionals, including trainees, it is critical to identify potential solutions to this growing public health crisis. In all, a more resilient workforce is one that would potentially rebound from stressful experiences with ample support and leadership focused on this need. The American Psychological Association defines resilience as “the process and outcome osuccessfully adapting to difficult or challenging life experiences, especially through mental, emotional, and behavioral flexibility and adjustment to external and internal demands” [15]. Factors contributing to the development of resilience include the availability and quality of social resources and specific coping strategies. The degree of resilience an individual experiences at any time can be measured through the Brief Resilience Scale [16], querying generally about the rapidity with which an individual can “snap back” after a stressful event. Resilience represents the convergence of a variety of psychological factors [17]. Specific individual qualities associated with personal resilience include self-awareness, emotional presence, playfulness, optimism, uncertainty tolerance, openness to collaboration, and hopefulness [18,19,20]. Potential interventions have included workplace programs such as education about burnout, workload modifications, increasing the diversity of work duties, stress management training, mentoring, emotional intelligence training, and wellness workshops. Interventions at the individual level have included promoting interpersonal professional relations, meditation, counseling, and exercise—all of which have been shown to be helpful [21].

Additionally, systems can be seen as resilient, impacting the very individuals working within them. As health care providers came to learn during the COVID pandemic, those health care systems that were able to maneuver to work with the new stress of the unknown catastrophic experience facilitated a greater sense of collaboration from those providing care. These systems generally demonstrate three components of resilience: the ability to absorb, adapt, recover, and transform in the face of stressful circumstances. Individual health care workers themselves on the front line must be a part of this adaptation through organizational advocacy to sustain a sense of meaning and resilience in the face of adversity [22].

In March 2022, we developed the Behavioral Health Providers Workforce Resiliency (BHPWR) ECHO Program to support health care providers by developing both skills in the development of resilience as well as a community of connection and social support (iecho.org/echo-institute-programs/supporting-resilience-through-health-care). This program included education on the components of resilience in the face of stress, education about specific resilience skills, presentations from experts around the nation, cases for discussion, and breakout rooms termed “resilience conversations” for participants to build connections and share information. Presentations included both individual sources of resilience and an understanding of how to impact the systems in which providers worked.

The objective of this analysis is to understand the impact of the BHPWR ECHO Program on the resilience and burnout of participating behavioral health professionals.

## 2. Materials and Methods

### 2.1. Development of the Program and Curriculum

Project ECHO is a novel tele-mentoring hub and spoke model for sharing knowledge between health care providers using an “All Teach, All Learn” approach. The model uses a mix of educational didactics and case-based learning to create a community of practice and improve care in underserved areas.

The origin of the BHPWR ECHO Program began in 2019 as a program for first responders struggling with burnout amidst an epidemic of opioid overdose deaths [23]. After the COVID-19 pandemic began and the call was opened internationally to all health care providers, it became apparent that health care professionals were calling in for connection and community amidst an uncertain, disconnected, isolating, and stressful time [24]. In response, the ECHO format was shifted from case-based learning to one involving short didactic sessions with listening groups. A new program, the BHPWR ECHO Program, was developed—aimed at first for behavioral health providers. The BHPWR ECHO Program includes three session components. The BHPWR Rounds ECHO, which includes a 40 min didactic followed by 20 min of question, answer, and discussion, started on 28 March 2022. This was followed by BHPWR Conversations (formerly called Workgroups) ECHO, which is a brief didactic followed by small-group breakouts for participants to share experiences and best practices, which began on April 25. Lastly, BHPWR Cases in Resilience ECHO (formerly Office Hours) was started in September 2022 and engaged participants in both real and simulated cases.

### 2.2. Attendance

BHPWR ECHO Program sessions were held virtually on Zoom [25], which tracked participant attendance. Participants were asked to register prior to the session and enter their name, e-mail, gender, age group, race/ethnicity, profession/discipline, credentials, and location. Providing this information was voluntary. Project ECHO staff members, the hub team, and guest presenters were excluded from the analysis. Attendance counts included both participants who attended by telephone and by video. No-cost continuing education credits were provided for each session. We assessed the number and percent of participants by characteristic and by session date.

### 2.3. Participant Survey

Program staff sent an annual online retrospective pre/post survey on two occasions during the program to any participant who had attended at least once: December 2022 and February 2024. If participants answered the survey on both occasions (n = 6), the 2024 results were used. The survey was administered using RedCap, a secure online survey application system [26], was administered to measure changes in participants’ self-perceived burnout and resilience. Survey topics included participants’ knowledge of managing workload, responding to a lack of control in the workplace, and responding to compassion fatigue. The survey also asked about participants’ confidence in building a support system, using the resiliency tools covered in the BHPWR ECHO Program curriculum to build self-resilience and resilience among the people that participants serve. The BHPWR ECHO Program is an “open cohort” program, meaning that participants can begin and end participation at their discretion. In order to best assess participants’ perceived changes in knowledge and confidence, we conducted a retrospective pre/post analysis. Participants were asked to rate their agreement with knowledge and confidence statements measured on a 5-point Likert scale (Strongly disagree, Disagree, Neutral, Agree, Strongly Agree). Participants were also asked to rate their level of professional burnout and resilience on a continuous slider scale from 0 to 50. For burnout: 0 = “I am not at all burnt out”, 25 = “I am feeling a little burnt out”, and 50 = “I have difficulty completing tasks because I am very burnt out”. For resilience: 0 = “I have great difficulty navigating and/or recovering from challenges”, 25 = “I am sometimes able to navigate and/or recover from challenges”, and 50 = “I am always able to successfully navigate and/or recover from challenges”. Mean scores were calculated for retrospective pre- (respondents were asked after they had already attended, but they were asked to think back to before they attended the ECHO) and post-responses and compared using a Wilcoxon signed ranks test for dependent ordinal/non-normally distributed variables.

### 2.4. Focus Interviews

Four focus interviews total were conducted virtually at two time points during the program, December 2022 and February 2024. Any participant who had attended at least one session was invited to participate. Discussions were recorded, transcribed, and then analyzed with NVivo 14, a qualitative coding and analytic software [27]. Transcripts were inductively and independently coded by two trained evaluation staff. Evaluation staff reviewed codes, came to consensus coding, sorted codes into categories, and identified major themes [28]. Focus group questions included the following: (1) What do you feel are the greatest sources of burnout in your field? In what ways did the ECHO support or not support reducing these sources of burnout? (2) What are the greatest sources of resilience in your field? In what ways did the Behavioral Health Professional Workforce Resilience ECHO support or not support these other resilience activities? (3) In which ways do you feel you have control over sources of burnout in your workplace? In what ways did the ECHO help or not help you feel like you had more control? Or that you could show others how to have more control over the sources of burnout in their lives? (4) What were the greatest benefits of attending the ECHO? (5) In what ways have you applied (or not applied) the tools you’ve learned in the ECHO? In your personal life? In your professional life? (6) Which sessions did you find most useful? What was it about these sessions that really stood out for you? (7) How has this program helped or not helped to create a community of practice? (8) For those of you that also attended all three ECHO programs, how did the ECHOs compare? In what ways did they complement each other? In what ways were they redundant? and (9) what can we do to improve this program? What do you wish the facilitators had done differently?

This study was approved by the Institutional Review Board (IRB) of the University of New Mexico Health Sciences Center (#22-216 Behavioral Health Workforce Resilience ECHO Evaluation).

## 3. Results

### 3.1. Program

From March 2022 through March 2024, there were 92 BHPWR ECHO Program sessions (BHPWR Conversations ECHO: 51, BHPWR Rounds ECHO: 27, and BHPWR Cases in Resilience ECHO: 14). Topics ranged from more conceptual topics, e.g., Motivation and Growth Mindset, to more skill-based sessions, e.g., Sleep and How to Optimize it for Resilience and Self-Care (see Table 1). Sessions focusing on the individual were also intentionally interspersed with systems-level sessions (e.g., Getting Your Leadership Onboard to Promote Wellness). The first months of the program focused on internal sources of resilience, including issues such as self-care. Further months have focused on stressors within health care systems, including those presented by differing backgrounds and perspectives, racial inequities, administrative burdens, evolving technologies, and the lack of trust within health care relationships. As the program continued, a diverse scope of health professionals providing behavioral health services (nurses, primary care physicians, counselors and social workers, etc.) began attending the program, resulting in the program being opened to any health care professional interested in developing resilience supports.

### 3.2. Attendance

The Behavioral Health Providers Workforce Resiliency (BHPWR) ECHO Program had 1585 people attend at least one session. Attendance was highest in the earliest part of the program and then stabilized to between 50 and 100 participants per session beginning in May 2022 (see Figure 1). Participants attended five sessions on average.

Among participants who reported demographics, women were much more likely to attend (see Table 2). This was particularly the case for the BHPWR Rounds ECHO. Participants were more likely to be older (over 40). Among participants who reported race/ethnicity, white was the most commonly reported race. Non-behavioral health professionals made up the largest percentage of participants. Among behavioral health professions, clinical mental health counseling and health care administrators were the most common participants. A total of 45 of 50 states were represented in the participation—the states that did not have participation were Arkansas, Tennessee, West Virginia, Delaware, and New Hampshire. Demographic questions at registration were voluntary. Therefore, the largest percentage for most demographic characteristics was “missing”.

A total of 53 people responded, which was 3.3% of the total participants and 57.6% of the 92 people who attended BHPWR ECHO Program sessions when the survey was active (6–19 December 2022 and 5–19 February 2024). Survey respondents reported that they perceived significant increases in all six measures of knowledge and confidence compared to when they started attending the program (see Table 3). Participants reported increased knowledge of how to respond when workloads became unmanageable, when feeling they lacked control in their life, when work felt less rewarding, and when experiencing compassion fatigue. They also reported increased confidence in building a good support system if needed, the ability to use the wellness tools that they learned in the course, and confidence to use the tools learned in the course to improve their patients’ wellness. Additionally, survey respondents reported feeling less burnt out professionally and more resilient against stressors in the workplace compared to when they started attending the ECHO program.

### 3.3. Focus Interviews

Nine participants attended the focus groups. There was participation from each region of the United States: Northwest, Northeast, Midwest, Southeast, and the West Coast. Additionally, one participant was based internationally. One participant was a health educator, one participant was in public health, two participants were clinical mental health counselors, one was a clinical psychologist, one was the director of a behavioral health non-profit, one was an obstetrician/gynecologist, and two did not state their profession. Each shared thoughts, including (1) sources of burnout, (2) sources of resilience, and (3) how the ECHO did or did not support their resilience and reduced their burnout.

All of the focus group participants reported many sources of burnout for health care professionals providing behavioral health treatment. Major themes that emerged were the patient load, lack of effective leadership, low pay, dehumanization of the profession, overwork, and the nature of providing behavioral health services. The dehumanization of the profession referred to how focus group participants felt that they were pressured to see more and more patients and dispense medications without truly checking in with their patients’ emotional needs. For example, one participant stated, “*…the conveyor belt of patients. It’s just back-to-back, back-to-back, back-to-back. More and more time required to spend charting when the patient is in the room. And that is not how I was trained in terms of traditional psychotherapy*”. They also felt that workplace administration contributed to burnout. For example, one participant stated, “*So, the inconsistent schedules, the lack of sleep, and the unsafe working environment, not just staffing purposes and what not, and feeling exhausted, the inability of management, or team leaders to foster emotional safety is a huge source of burnout*”. Lastly, participants reported sometimes feeling unsafe in the workplace. This was particularly true for professionals who worked in in-patient or incarceration settings: “*In any inpatient unit you could have a crisis situation*”.

Focus group participants reported a number of factors that contributed to the resilience of behavioral health care professionals. Many participants reported leaning on their friends and family, having good leadership, and engaging in self-care as much as possible (eating well, exercising). One participant stated, “*You need to have someone who loves you*”. Focus group participants reported frequently leaning on their colleagues to help them when their work became stressful, mostly just to have someone who understood their work listen to them. One participant stated, “*Being able to go into someone else’s office at the end of the day. Just when you thought you couldn’t hear any more stories—awful stories—you could go in and kind of unload before you go home that night*”. One participant expressed that the call to serve was, in itself, a source of resilience. They felt that helping a patient allowed them to remain stronger and more resilient personally. One participant stated, “*It’s an honor. It’s a privilege to actually help someone…It’s an honor to save somebody’s life*”.

When asked about the ECHO program specifically, focus group participants combined the concepts of burnout and resilience in their responses, so the two were combined in the analysis. Participants felt that one source of the ECHO’s effectiveness was its interdisciplinary community. Participants liked that attendees and the hub team members came from different fields/professions and that this provided different viewpoints on the topics being covered.

One participant stated, “*When they opened up to more types of providers … that was wonderful*”. Participants also appreciated that the ECHO included a connection to systems change. Several participants reported that learning self-care exclusively, when the source of their burnout was at a systems level, was frustrating. One participant stated, “*Sometimes they’ll have these resiliency programs and they want you to, you know, somehow if you do your mindfulness practice that’s going to solve it all and the system doesn’t have to change. That always makes me kind of resentful*”. They liked the resources that were provided, and one participant reported that she saved every presentation and referred back to the slides in her own work. Participants liked the program structure and felt that the three ECHOs complemented each other, and several provided specific instances of applying the content to their lives. Lastly, focus group participants reported that they felt that the ECHO successfully built a community of people providing behavioral health care. One participant stated, “*It’s nice to be with a group of people that wants to do good work and wants to continue to do it. To find a way—to not quit. Just figure it out. There’s got to be a way that we can keep doing this difficult work during this really unpleasant time without not falling apart and losing everything*”. Another participant reported, “*What I found really helpful about the ECHO was that especially the resilience, ECHO, because yeah, you get the information and everything but it also gave names to what I was experiencing… and then having experts in the field like offer suggestions to help counteract, that was also helpful*”.

## 4. Discussion

The Behavioral Health Providers Workforce Resiliency (BHPWR) ECHO Program has reached providers across the nation and multiple countries outside of the United States. Self-reported burnout risk was reduced among participants, and the data support the program’s preventive impact.

Participants in the program expressed gratitude for the ability to participate in online chats, in session feedback, in program reviews, and in listening groups. Of great importance, participants subjectively experienced a decreased level of burnout and enhanced professional fulfillment in correlation with participation in this program. In all forms of program feedback, participants expressed that the connections they experienced in focus groups were the most important component of the program, though many expressed the great value in the didactic information presented. Many reported sharing this information with others in their institution.

While the program was originally aimed to support behavioral health clinicians exclusively, it became clear that many participating in the program were health care professionals from all branches of medicine and multiple disciplines. It became apparent that individuals across health care are seeking support to help them care for others, so the program was opened up to all health care professionals. We have learned that health care professionals further along in their careers appear to have more time available to participate in the program. While research has shown that students and trainees express high levels of burnout and even suicidal ideation across medical fields—reaching 50% in medical school [29] and averaging 35% in residency training programs [30]—busy schedules make it difficult to find time and coverage to participate. The desire for support is high, but when it comes to the actual available hours, busy health care professionals are more likely to turn to catching up on tasks to reduce stress than to schedule themselves into a supportive program. Given the scope of provider burnout and anticipated departure from health care that looms as a public health crisis, the lack of ability to make it to a desired program is of grave concern.

Professional burnout has become a critical public health crisis. According to a recent survey, 20% of nurses and 40% of physicians say it is likely they will leave their current work within the next two years, and one-third of health professionals say that it is likely they will reduce their hours within the next twelve months [31]. A survey of 20,000 professionals at 124 institutions cited burnout, excessive workload, fear of infection, emotional issues from COVID-19 work, and the number of years in practice as contributing to this “great resignation” within medicine. The study suggested that a potential solution to this issue involved a focus on making employees feel valued at work and establishing emotional support networks [32].

While studies of health care professionals indicate that they are actually more resilient than matched employees in fields outside of medicine [33], the stresses of medicine these days call for building even higher levels of resilience within the clinical workforce. We have learned the complicated nature of resilience through feedback in this program. A curriculum must present ideas that an individual can implement in their own life—internal sources of resilience. A resilience program must also address the complicated system in which health care providers work and the multiple sources of burnout that the system potentially produces. Addressing efficiencies of practice and the culture in which we work is essential to improving health care worker wellbeing. While an individual can develop skills and connections and consider forces in their own personal lives that detract from the experience of resilience, the system in which a provider finds themselves is of equal importance. With additional stressors, systems must be able to absorb and adapt to new inputs and transform as health care organizations. Listening groups allow time for professionals to discuss what they have done within their own systems to address these challenges, including issues of loneliness, structural racism, lack of diversity, and the burdens presented through inefficient workflows. A strong theme emerging both in the literature of resilient systems and from the minds of our participants involves working with individuals on the frontlines to provide health care in the identification of systemic needs.

Ultimately, health care providers are the centerpiece of any health care system, and their personal welfare must be understood. Our experience with this program underscores the idea that working with difficult feelings through the everyday language of human interactions in online forums is highly appreciated by individuals providing care. Providing a space for the expression of these human experiences amidst difficult circumstances has the potential to assuage the experience of existential anxiety [34]. However, though resilience programs in and of themselves can go a long way to support the welfare of those providing care, those in charge of funding these organizations, whether executives, legislators, or board members, must consider the impact of systems that may impact the welfare of employees charged with providing the care for all of us. This issue is identified in multiple international studies of health care systems. One of particular note, given the adversity under which health care providers currently provide care, involves the crisis in Sudan. The resilient health care system there has been able, in some ways, to consider critical components, including absorbing, adapting, and transforming in response to catastrophic stressors [22]. Studies examining critical issues for health care provider resilience across the globe in the context of the COVID-19 crisis include the protection and experiences of frontline health care workers, the role of health systems and policy, planning and management issues, and education and health labor markets [35]. 

Nevertheless, we are hopeful that this model will be an important model to support health care workers through the stresses encountered in their work, with the potential for replication worldwide for those who spend their lives in the service of caring for others.

There are limitations that should be considered when interpreting these results. The primary limitation is that participation in the assessment of the program was completely voluntary. Therefore, information was not available on a large number of participants and potentially a biased group of participants. Participants in the program may be more likely to be non-white race-ethnicities compared to the overall health care population. For example, the Association of American Medical Colleges reports that 56.5% of physicians are white, whereas only 29.2% of ECHO participants are white. Additionally, resources were not available to assess more long-term outcomes of resilience and burnout (e.g., high turnover, mental health diagnoses, etc.) [36].

There is a need to support health care professionals who provide critical health care services, whether medical or behavioral health. Virtual, tele-mentoring programs like the Behavioral Health Providers Workforce Resiliency (BHPWR) ECHO Program can efficiently and effectively improve wellness and mental health among health care professionals.

## 5. Conclusions

The Project ECHO tele-mentoring program, which focuses on building resilience in clinicians, shows great promise. It is potentially reproducible to help support individual providers across a large organization or in another nation, replicating the use of virtual connections. This particular program involved a mix of didactic presentations, case discussions, and participant engagement through listening groups. The program evaluations have demonstrated that the program is helpful for clinicians across all fields of medicine and for all disciplines of clinical providers. Our experiences have shown us that a critical component of this program involves the creation of listening groups through breakout rooms. Our hub team providers were quite experienced in running groups in various formats and felt comfortable working with individuals unknown to them about a variety of topics, ranging from skill-building exercises to sharing experiences. Though rarely occurring, it was important that our providers had the sensitivity to know how and when to reach out to participants if they felt there may be a need for greater conversation, perhaps with an individual therapist. It will be noted, however, that despite the tremendous adversity expressed by some participants in their locations, the need for further contact with participants was rare. Rather, participants expressed appreciation for the opportunity to share their experiences. We believe that any organization attempting to reproduce this program should consider the role of mentorship in program development so that the format and content align with the specific needs of the participants involved.

An emphasis on burnout prevention and the consideration of components of resilience is key to all health care organizations. At this time, health care is in crisis, with providers from almost all disciplines reconsidering their desire to practice in the future, representing a true threat to the capacity of the U.S. health care system. Resilience efforts cannot come from a single class or an insistence that providers contemplate the ideas of “work-life balance”. Rather, leaders in an organization can consider how an ongoing program fostering connections among providers can build a resilient workforce. Simultaneously, those involved in the administration of health care must care about those factors that can augment the resilience of a system, allowing it to absorb stresses and transform models of care. Simultaneously, those providing direct care must continually have input as to how current administrative burdens can be streamlined to support the relationship between clinicians and their patients.

## Figures and Tables

**Figure 1 healthcare-12-01741-f001:**
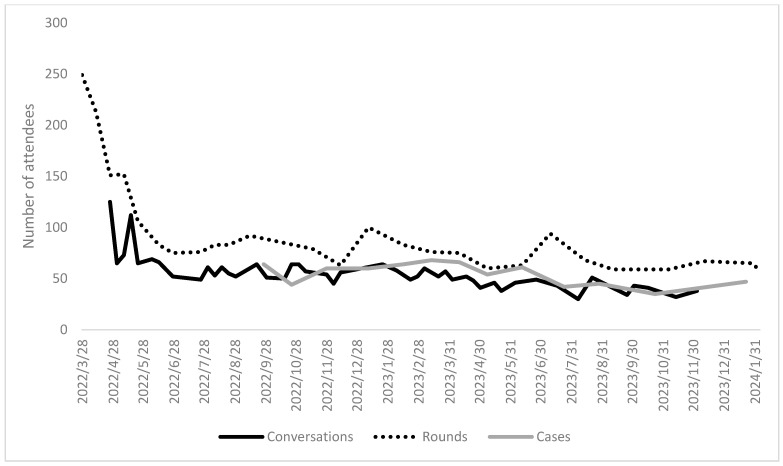
Participant attendance by ECHO, BHPWR ECHO Program, 28 March 2022 to 25 March 2024, N = 1585.

**Table 1 healthcare-12-01741-t001:** Examples of session topics for the Behavioral Health Providers Workforce Resiliency (BHPWR) ECHO Program by ECHO, March 2022–March 2024.

BHPWR Conversations ECHO	BHPWR Rounds ECHO	BHPWR Cases in Resilience ECHO
Introduction to the Peak Performance Series	Introduction to the Program	Sleep & How to Optimize it for Resilience & Self-Care
Self-Awareness	Professional Performance and Burnout	
Self-Care	Resilience and Training	Introduction to Cases in Resilience: Developing resilience and wellness interventions for impaired staff/employees: Part 1
Motivation and Growth Mindset	Peer Support Networks	
Relationships	Intergenerational Trauma and Resilience	
Perspective Taking	Community Resilience Model (CRM)	
Uncertainty, Tolerance, and Flexibility	Qualities of Resilience	Case I: PT II Supporting organization after an impaired employee intervention. Organizational and employee resilience and wellness development.
Values and Character	Using Supervision to Improve Resilience	
Introduction to Series 2-The importance of connection	Getting Your Leadership Onboard to Promote Wellness	
The Power Pause	Trauma & Resilience	
Post Traumatic Growth	Flexibility and Resilience	
Peer Support for Clinicians	Uncertainty and Resiliency	Considering Personal Limits to Prevent Burnout: A case-based learning experience
Diversity, Equity, Inclusion, and Wellness Part 1	Nutrition & Resilience (Performance Nutrition)	
Diversity, Equity, Inclusion, and Wellness Part 2	Reorienting the US Healthcare System Around Relationships	Part 2 Case of Marie: Exploring Work Relationships
The Art of Saying No: Setting Boundaries and Finding Your Professional Voice	Supporting Your Resiliency by Reducing those ‘Pebbles in Your Shoes’	Age and Well Being: Resiliency for a Seasoned Provider
When it’s Time to Change	Burnout: What I Learned as AMA President	A Trainee Adapting to a Health Care Organization
Recognizing When We Need Help: Caring for the Caregivers	Addressing the Elephant in the Room: Shame and Sentinel Emotional Events in Medical Learners	Burnout: The Supervisory Impact of an Employee’s Resignation
Relationships Built on Trust: Creating and Developing Mentor-Mentee Relationships by Building Trust	The Public Health Epidemics of Substance Use, Chronic Pain, Mental Health, and Suicide: How can we overcome these crises with resilience?	A Tale of Two Employees: Navigating Disruptions in Supervisory Relationships
When a Patient Dies by Suicide	Mitigating Clinician Burnout: The Strategic Role of Leadership Support in Promoting a Culture of Well-Being	The Resilient Supervisor
Psychological Impact of a Lawsuit	The Power of Play: Incorporating Fun into Your Busy Schedule: The Antidote to Burnout	Making Mistakes/Perfectionism
Peak Performance: Self Awareness	Improv4Health Professionals: Taming Your Inner Critic	Empathy and Flexibility of Mind
Peak Performance: Self-Regulation and Mental Flexibility	Safeguarding Our Resilience and Wellbeing as Care Givers: Lessons from the Hidden Culture of 911 Professionals	The Open Source Healing Initiative
Peak Performance: Sleep Hygiene	Helping Those Who Help Nurture and Maintain Their Resilience	
Peak Performance: Building Resilient Teams	Trauma and Community Resiliency Model	
Relationships Built on Trust: Trust between Medical Specialties		
Building Trust with Refugee Families in a Multidisciplinary Pediatric Clinic		
Relationships Built on Trust: Trust between Behavioral Health Professionals and Care Providers in an Integrated Clinic		
How to Make Work Your Playground; Developing and Building Trusting Relationships Across Campus		
Relationships Built on Trust: A Primer on Transformative Justice		

**Table 2 healthcare-12-01741-t002:** Characteristics of participants in the BHPWR ECHO Program by ECHO and program year ^1^, 28 March 2022 to 25 March 2024, N = 1585.

Gender	Total N = 6027, n (%)		Rounds N = 2516, n (%)		Conversations N = 2752, n (%)		Cases N = 759, n (%)	
	Year 1	Year 2	Year 1	Year 2	Year 1	Year 2	Year 1	Year 2
Female	1337 (33.1)	1187 (59.7)	698 (68.0)	223 (60.8)	296 (43.3)	153 (55.6)	106 (47.1)	128 (56.9)
Male	523 (12.9)	398 (20.0)	82 (8.0)	55 (15.0)	76 (11.1)	50 (18.2)	33 (14.7)	35 (15.6)
Non-binary ^2^	9 (0.2)	15 (0.8)	3 (0.3)	3 (0.5)	3 (0.3)	9 (3.3)	0 (0.0)	3 (1.3)
Missing	2173 (53.8)	389 (19.6)	243 (23.7)	87 (23.7)	309 (45.2)	63 (22.9)	110 (48.9)	59 (26.2)
**Age Group (years)**								
≤19	0 (0.0)	5 (0.3)	0 (0.0)	0 (0.0)	0 (0.0)	5 (1.8)	0 (0.0)	0 (0.0)
20–29	25 (0.6)	38 (1.9)	14 (1.4)	6 (1.8)	16 (2.3)	8 (2.9)	2 (0.9)	4 (1.8)
30–39	122 (3.0)	86 (4.3)	57 (5.6)	18 (4.9)	38 (5.6)	12 (4.4)	13 (5.8)	17 (7.6)
40–49	572 (14.2)	311 (15.7)	460 (44.8)	69 (18.8)	79 (11.6)	44 (16.0)	29 (12.9)	38 (17.0)
50–59	205 (5.1)	178 (9.0)	80 (7.8)	58 (15.8)	62 (62)	28 (10.2)	20 (8.9)	20 (9.0)
≥60	754 (18.7)	488 (24.6)	119 (11.6)	62 (16.9)	136 (19.9)	66 (24.0)	51 (22.7)	46 (20.6)
Missing	2364 (58.5)	879 (44.3)	296 (28.8)	154 (45.0)	352 (51.5)	112 (40.7)	110 (48.9)	98 (43.9)
**Hispanic Identity**								
Yes	279 (6.9)	362 (18.2)	32 (3.1)	44 (12.0)	52 (7.6)	42 (15.3)	20 (8.9)	21 (9.4)
No	1773 (43.9)	1282 (64.5)	141 (13.7)	152 (41.4)	243 (35.6)	184 (66.9)	82 (36.4)	102 (45.7)
Missing	1990 (49.2)	345 (17.3)	853 (83.1)	171 (46.6)	388 (56.8)	49 (17.8)	123 (54.7)	100 (44.8)
**Race**								
AI/AN	127 (3.1)	86 (4.3)	33 (3.2)	10 (2.7)	22 (3.2)	6 (2.2)	8 (3.6)	7 (3.1)
Asian	161 (4.0)	119 (6.0)	9 (0.9)	10 (2.7)	20 (2.9)	16 (5.8)	3 (1.3)	6 (2.7)
Black/AA	111 (2.7)	104 (5.2)	22 (2.1)	15 (4.1)	18 (2.6)	15 (5.5)	11 (4.9)	14 (6.3)
NHOPI	0 (0.0)	1 (0.0)	0 (0.0)	0 (0.0)	1 (0.1)	0 (0.0)	0 (0.0)	0 (0.0)
Hispanic ^2^	132 (3.3)	116 (5.8)	37 (3.6)	15 (4.1)	36 (5.3)	16 (5.8)	9 (4.0)	9 (4.0)
White	1180 (29.2)	963 (48.5)	217 (21.2)	144 (39.2)	241 (35.3)	139 (50.5)	69 (30.7)	81 (36.3)
More than one race	203 (5.0)	165 (8.3)	45 (4.4)	14 (3.8)	40 (5.9)	27 (9.8)	8 (3.6)	8 (3.6)
Prefer not to answer	132 (3.3)	72 (3.6)	7 (0.7)	7 (1.9)	25 (3.7)	9 (3.3)	6 (2.7)	6 (2.7)
Missing	1996 (49.4)	360 (18.1)	656 (63.9)	150 (40.9)	280 (41.0)	47 (17.1)	111 (49.3)	92 (41.3)
**Profession**								
Non-behavioral health	606 (15.0)	596 (30.0)	519 (50.6)	95 (25.9)	133 (19.5)	90 (32.7)	36 (16.1)	47 (21.1)
Administrator/program manager	291 (7.2)	302 (15.2)	32 (3.1)	36 (9.8)	67 (9.8)	42 (15.3)	8 (3.6)	18 (8.1)
Clinical mental health counseling	326 (8.1)	236 (11.9)	53 (5.2)	28 (7.6)	44 (6.4)	30 (10.9)	19 (8.5)	17 (7.6)
Clinical psychology	140 (3.5)	102 (5.1)	38 (3.7)	15 (4.1)	32 (4.7)	11 (4.0)	15 (6.7)	11 (4.9)
Psychiatrist	258 (6.4)	171 (8.6)	19 (1.9)	17 (4.6)	30 (4.4)	20 (7.3)	14 (6.3)	16 (7.2)
Public health social work	58 (1.4)	12 (0.6)	42 (4.1)	6 (1.6)	29 (4.2)	6 (2.2)	4 (1.8)	4 (1.8)
Clinical social work	202 (5.0)	87 (4.4)	18 (1.8)	17 (4.6)	26 (3.8)	9 (3.3)	12 (5.4)	9 (4.0)
Counseling psychology	42 (1.0)	32 (1.6)	19 (1.9)	7 (1.9)	19 (2.8)	3 (1.1)	7 (3.1)	7 (3.1)
First responder	31 (0.8)	20 (1.0)	6 (0.6)	6 (1.6)	9 (1.3)	6 (2.2)	0 (0.0)	2 (0.9)
Other ^3^	36 (0.9)	60 (3.0)	8 (0.8)	10 (2.7)	23 (3.4)	12 (4.4)	2 (0.9)	7 (3.1)
Missing	2052 (50.7)	367 (18.5)	272 (26.5)	130 (35.4)	271 (39.7)	46 (16.7)	106 (47.5)	85 (38.1)

^1^ Year 1 = March 2022 to March 2023; Year 2 = April 2023 to March 2024; programs are not mutually exclusive, so program numbers will not sum to total numbers. Total numbers are total unique participants who have attended at least one ECHO. ^2^ Non-binary or non-conforming. ^3^ Care coordinator, retired, student, community health worker, quality improvement, substance use disorder prevention.

**Table 3 healthcare-12-01741-t003:** Participant responses to retrospective pre/post survey for BHPWR ECHO Program, 28 March 2022 to 25 March 2024, N = 53.

Changes in Knowledge and Confidence ^1^	Before Score, Mean (SE)	After Score, Mean (SE)	*p*-Value
I know how to respond when my workload was more than I could manage	3.5 (0.2)	4.4 (0.1)	<0.001
I know how to respond when I felt I lack of control in my life	3.5 (0.1)	4.3 (0.1)	<0.001
I know how to respond when my work did not feel rewarding	3.2 (0.2)	4.2 (0.1)	<0.001
I know how to respond when I felt like I had less compassion for the people I work with (compassion fatigue)	3.2 (0.2)	4.2 (0.1)	<0.001
I feel confident that I could build a good support system if needed	3.4 (0.2)	4.4 (0.1)	<0.001
I am confident that I could use tools (community resilience model, resilient parenting, etc.) to build my own resiliency	3.2 (0.3)	4.4 (0.1)	<0.001
I am confident that I could use tools (community resilience model, resilient parenting, etc.) to build resiliency in the people I serve (e.g., patients, students, employees, etc.)	3.0 (0.2)	4.2 (0.1)	<0.001
**Changes in outcomes ^2^**			
How would you rate your level of feeling burnt out professionally?	26.0 (1.9)	17.8 (1.7)	<0.001
How would you rate your level of being able to protect yourself against the stressors in the workplace (i.e., resilience) in general?	29.9 (1.7)	34.9 (1.5)	0.005

^1^ Knowledge and confidence statements measured on a 5-point Likert scale (strongly disagree = 1, disagree = 2, neutral = 3, agree = 4, strongly agree = 5). ^2^ Participants were also asked to rate their level of professional burnout and resilience on a continuous slider scale from 0 to 50. For burnout: 0 = “I am not at all burnt out”, 25 = “I am feeling a little burnt out”, and 50 = “I have difficulty completing tasks because I am very burnt out”. For resilience: 0 = “I have great difficulty navigating and/or recovering from challenges”, 25 = “I am sometimes able to navigate and/or recover from challenges”, and 50 = “I am always able to successfully navigate and/or recover from challenges”.

## Data Availability

The data and materials from this manuscript are not publicly available.

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
