# Peer review of "Caring for the Caregivers: Improving Mental Health among Health Professionals Using the Behavioral Health Professional Workforce Resilience ECHO Program"

_healthcare, 2024, doi:10.3390/healthcare12171741_

Round 1

Reviewer 1 Report

Comments and Suggestions for Authors

Review for healthcare (MDPI)

Title: Caring for the Caregivers: Improving Mental Health among Behavioral Health Professionals Using the Behavioral Health Professional Workforce Resilience ECHO Program

Thank you for the opportunity to read and comment on this interesting article.

  • Relevance: The topic of preventing burnout among behavioral health professionals is highly relevant.
  • Methodology: The study uses a solid approach with a small sample size and mixed methods.

Title: I wonder if it should be “Improving Mental Health among Health Professionals,” since you mentioned growing interest in the program from professionals outside the mental health field.

Abstract: The abstract gives good information about background, methods and results. The number of N=1,532 participants in the program is slightly misleading and can be misunderstood as the number of participants in the study, which is N=62 – it might be helpful to clarify this.

Introduction: The introduction clearly defines burnout and provides relevant statistics on burnout rates and intent to leave, which underscores the study's importance.

Materials and Methods: This section includes everything necessary to understand the study's design. For better readability, if the journal allows, I suggest presenting the survey/focus group questions in list form and providing a visual representation of the pre-post questioning timeline. The current format is slightly confusing.

Table 1: The chronological order of the content confused me (Rounds: March 2022, Conversations: April 2022, Resilience: September 2022). Additionally, consider reducing the table to a few examples that showcase the variety of topics; the full list of titles can be overwhelming.

Participant Demographics: Information on participants' gender and ethnicity is interesting, but it would be more meaningful if you compared it to the general distribution in the healthcare workforce. Is the distribution representative?

Survey Participation: A participation rate of 3.3% seems problematic. Why was the survey only open during two short periods?

Focus Groups: The mixture of participants is good. However, with only 9 participants spread across 4 focus groups, I question the definition of a “group.” Please specify or consider renaming them to “focus interviews with 1-3 participants.”

Quotes: The collection of quotes is well-structured. However, the last quote lacks italicization.

Discussion:

  • “The program has successfully reduced self-reported burnout and increased resiliency” is a strong statement that doesn’t seem fully supported by the data. There is no report on dropout rates or whether the burnout levels of participants were representative. It is possible that those who were already experiencing burnout were less likely to participate. You also mention that students and trainees are at higher risk for burnout but are less likely to participate. Therefore, it might be safer to say that self-reported burnout risk was reduced among participants and that the data support the program's preventive effect.
  • The statement that “the connections they experienced in focus groups were the most important component of the program” is problematic. There were only 9 participants in the focus groups, and the focus groups were part of the program evaluation, not the program itself. If the “rounds” of the program are also called “focus groups,” this needs clarification.
  • “A resilience program must also address the complicated system in which healthcare providers work and the multiple sources of burnout that the system potentially produces” is a crucial point. It refers to a participant's quote about the frustration that arises when “resilience and mindfulness programs shall solve it all.” I encourage you to rephrase this to emphasize that those in power—politicians and other stakeholders—hold the responsibility to change the system. Expecting too much from a resilience program could be unrealistic. The program can address symptoms and reduce consequences, and it might contribute to systemic change, but it cannot be the change itself. I recommend discussing:
    • Binder, P.-E. (2022). Suffering a healthy life—on the existential dimension of health. Frontiers in Psychology, 13, 803792. doi:https://doi.org/10.3389/fpsyg.2022.803792.

Limitations: A paragraph on limitations is missing.

Conclusion: You don’t need the word “quite.”

Funding: Is there a word missing in the sentence, possibly “years”?

All in all, I enjoyed reading the article and I think the program is very encouraging.

Author Response

Please see attached Word document response.

Reviewer 2 Report

Comments and Suggestions for Authors

Dear authors. I have found reading your original work highly interesting and I truly think this is a timely and enriching contribution. I list below some points to take into consideration, with the sole aim of trying to improve the original which, in my opinion, would be publishable with minor changes.

The Introduction refers to some aspects of burnout, specifically in the field of health and medical education. However, it seems to me to be a relatively poor and hasty theoretical framework. The topic of study should be much more focused, and in particular, the relationships between the work context and burnout, which, in principle, do not seem very clear in the text. The Behavioral Health Providers Workforce Resiliency (BHPWR) ECHO apparently very interesting, would also deserve a more detailed explanation.

In the Discussion section there are very few up to date international references, that compare the results obtained in the context described in the research with other contexts and other research from the international scientific community. I sincerely believe that your work would gain a lot if this part of the paper were improved.

After a study that, in my opinion, is interesting and current, the conclusions seem somewhat poor. They could be rewritten to highlight, in a more forceful way, the main findings of the study.

It would also be interesting to add a clear explanation of the limitations of the study, especially its representativeness with respect to the general population under study (it seems that the sample is small compared to the population, this issue would have to be justified a little), as well as a statement of future research perspectives that this research opens up.

Kind regards

Author Response

(The authors gave the same response as above.)

Reviewer 3 Report

Comments and Suggestions for Authors

First, I would like to thank you for the opportunity to read this work. This study provides valuable information concerning using a Behavioral Health Professional Workforce Resilience ECHO Program among Behavioral Health Professionals. Thus, this study sheds light on the success of a program aiming to improve wellness among healthcare professionals. This kind of data is scarce; as such, I congratulate the authors for conducting this research, which has important implications for the field.

Below, I leave some comments, hoping they improve the paper.

Abstract: Although the abstract generally contains all the necessary information, the authors need to clarify the study's main findings when they start discussing them. In other words, I recommend not starting by saying, “Participants increased…”. Clearly highlight that you are going to start talking about the main findings. In addition, make the relevance of your study clear by highlighting its potential implications.

Introduction: The authors need to deepen the literature review and address the concept of resilience, as it is one of the constructs analysed. In addition, it is unclear why the authors addressed individuals in medical training and the issues concerning medical education in this section. It is important to make clear what population this study focuses on.

Material and Methods: There is a lack of scientific background concerning the development of the program and the participant's survey. In which theoretical background this program was anchored? In addition, concerning the instruments used to assess burnout and resilience, these measures were already used? Who are the authors? What are the psychometric properties of these instruments? For instance, for measuring burnout, I am concerned about the reliance on the items used, such as “I am not at all burnt out”. Are the participants aware of the criteria for diagnosing themselves with burnout? Why did the authors not use the Maslach Burnout Inventory? Or (although less used) the Shirom-Melamed Burnout Measure (SMBM)?

In addition, to allow the replication of the BHPWR ECHO Program in the future, it will be important to add more information, such as the authors, whether the program contents are available, and, if so, how to access the full description and contents of this program. The authors only added in the discussion: «The Behavioral Health Providers Workforce Resiliency (BHPWR) ECHO Program has reached providers across the nation and multiple countries outside of the United States and successfully reduced self-reported burnout and increased resiliency among professionals providing behavioral health care.» This information and references must have been added previously to the material and methods section.

Results: It will be important to add a potential explanation for the low response rate in the participant survey (3.3% of the total participants) and the focus group (why only 9 participants?).

Discussion: The authors defined resilience only in the discussion section. This definition must be included in the introduction.

Conclusion: It will be important to identify the potential limitations of this study and discuss potential implications. 

Author Response

(The authors gave the same response as above.)

Round 2

Reviewer 3 Report

Comments and Suggestions for Authors

The authors have addressed my previous comments and suggestions. I have nothing more substantial to add.